# Cluster-to-particle transition in atmospheric nanoclusters

Haide Wu, Yosef Knattrup, Andreas Buchgraitz Jensen, and Jonas Elm

Department of Chemistry, Aarhus University, Langelandsgade 140, 8000 Aarhus C, Denmark

**Correspondence:** Jonas Elm (jelm@chem.au.dk)

**Abstract.** The formation of molecular clusters is an imperative step leading to the formation of new aerosol particles in the atmosphere. However, the point at which a given assembly of molecules represent an atmospheric molecular cluster or a particle remains ambiguous. Applying quantum chemical calculations we elucidate this cluster-to-particle transition process in atmospherically relevant sulfuric acid–base clusters. We calculated accurate thermodynamic properties of large $(SA)_n(base)_n$ clusters ($n = 1 - 15$), with SA being sulfuric acid and the base being either ammonia (AM), methylamine (MA), dimethylamine (DMA) or trimethylamine (TMA). Based on our results, we deduce a property-based criteria for defining "freshly nucleated particles (FNPs)", that act as a boundary between discrete cluster configurations and large particles. We define the onset of FNPs as when one or more ions are fully solvated inside the cluster and when the gradient of the size-averaged binding free energy approaches zero. This definition easily allows the identification of FNPs and is applicable to particles of arbitrary chemical composition. For the $(SA)_n(base)_n$ clusters studied here the cluster-to-particle transition point occurs around 16–20 monomers.

We find that the formation of FNPs in the atmosphere depend highly on the cluster composition and atmospheric conditions. For instance, at low temperature (278.15 K) and high precursor concentration (AM = 10 ppb and MA = 10 ppt) the SA–AM and SA–MA systems can form clusters that grow to and likely beyond ~1.8 nm sizes. The SA–DMA system form clusters that grow to larger sizes at low temperature (278.15 K), independent of the concentration (DMA = $1 - 10$ ppt) and the SA–TMA system (1:1 acid–base ratio) can only form small clusters, that are unable to grow to larger sizes.

## 1 Introduction

The recent 2023 IPCC report verifies that aerosol-cloud interactions remain the largest uncertainty in global radiative forcing (Lee et al., 2023). New particle formation (NPF) processes are believed to account for up to half the amount of cloud condensation nuclei (Merikanto et al., 2009). The formation of new aerosol particles occurs through nucleation of gas phase vapours (Kulmala et al., 2013). Initially, small molecular clusters are formed via strong intermolecular interactions between atmospheric vapour molecules. Under the premise that these clusters do not evaporate or are lost due to coagulation with existing particles, they can grow to larger sizes, eventually becoming aerosol particles of over 2 nm in diameter, which is the detection limit of many standard atmospheric measurement sites, while smaller particles require mass spectrometer and condensation particle counter techniques.

Sulfuric acid (SA) has unequivocally been shown to be a prime driver of NPF over land and oceans (Sipilä et al., 2010). However, base molecules are required to facilitate the cluster formation process in the lower troposphere (Kirkby et al., 2011). Highly abundant bases with low basicity, such ammonia (AM), or less abundant, but more basic alkylamines such as methylamine (MA), dimethylamine (DMA) and trimethylamine (TMA) have been confirmed to participate in the cluster formation process. The seminal work by Kurtén et al. (2008) showed that the strong binding of less abundant, but stronger bases (such as DMA) could overshadow the effect of the more abundant, less strong bases such as ammonia in the initial clustering process with SA. This has been reaffirmed in the state-of-the-art CLOUD chamber experiments by Almeida et al. (2013).

Jen et al. (2014) studied the stabilization of sulfuric acid dimers by AM and alkylamines (MA, DMA and TMA) using a flow tube reactor setup. At SA concentrations in the range of $10^7$-$10^9$ molecules cm$^{-3}$ and base concentrations leading to saturated SA dimer concentrations, the following trend of dimer stabilization was identified: AM < MA < TMA ≤ DMA. Glasoe et al. (2015) expanded upon the work by Jen et al. (2014), by studying 1.8 nm sulfuric acid–base particle formation. Here an AM < MA < DMA < TMA trend was found. These studies imply that both the initial cluster formation rates and 1.8 nm particle formation rates are highly dependent on the basicity of the bases. Elm (2021a) studied small $(SA)_{1-2}(base)_{1-2}$ cluster formation using computational methods and found a similar trend in the cluster formation potential. Computational work by Kubečka et al. (2023b) has shown that TMA is involved in the initial SA–base cluster formation process, but larger clusters with more than 1-2 TMA molecules are unstable, making TMA evaporate. However, the presence of TMA could still contribute to NPF, increasing the nucleation rate by ∼50% at 298.15 K. This mechanism has been corroborated to occur in polluted environments, such as the urban Beijing atmosphere, where SA–base clusters with up to 1 TMA molecule were detected using a CI-APi-TOF (Cai et al., 2023). Composition-wise, both computational (Olenius et al., 2013; Elm, 2017) studies and experimental results (Kürten et al., 2014) have shown that SA–base cluster formation is most favourable, when there is a 1:1 ratio of the acids to bases, with the limiting step being the initial formation of the $(SA)_1(base)_1$ clusters (Elm, 2017; Cai et al., 2022)

Following the cluster formation process from single molecules up to measurable ∼2 nm particle sizes has previously not been directly possible for electrically neutral clusters using either experimental or computational techniques. Quantum chemical calculations can be applied to calculate accurate thermochemistry of the clustering process, thereby giving direct information about the relative importance of different clustering species. However, such accurate calculations are computationally expensive and can routinely only be performed on clusters containing up to a maximum of eight monomers (see recent comprehensive reviews (Elm et al., 2020, 2023; Engsvang et al., 2023b)). We recently pushed this limit by studying large $(SA)_n(AM)_n$ clusters, with $n = 1 - 30$ (Engsvang and Elm, 2022; Engsvang et al., 2023a). To study such large systems we had to reduce the level of theory and were limited to GFN1-xTB (Grimme et al., 2017) geometries and B97-3c (Brandenburg et al., 2018) single point energies. However, we identified large uncertainties in the calculated thermochemistry, which was ascribed to insufficient configurational sampling (Engsvang and Elm, 2022; Engsvang et al., 2023a). We recently addressed this issue and found that several parallel configurational sampling runs yielded the most reliable final configurations. Hence, we now have a computational methodology that can be applied all the way from single molecules to $(SA)_n(base)_n$ clusters up to ∼ 2 nm sizes.

Here, we further explore the chemical complexity of large clusters by studying $(SA)_n(base)_n$ clusters ($n = 1 - 15$), with SA being sulfuric acid and the base being either ammonia (AM), methylamine (MA), dimethylamine (DMA) or trimethy-

lamine (TMA). Based on the results we will deduce a property-based criteria for defining when a given assembly of molecules represents an atmospheric molecular cluster or a freshly nucleated particle (FNP).

## 2 Methods

### 2.1 Computational details

Density functional theory calculations employing the empirically corrected B97-3c (Brandenburg et al., 2018) method were performed in ORCA 5.0.4 (ORC). In the case of the $(SA)_{14}(TMA)_{14}$ system, we had to apply a VeryTightSCF criteria to ensure convergence. Semiempirical tight binding calculations were performed using the original GFN1-xTB (Grimme et al., 2017) model and a reparameterized GFN1-xTB model, denoted GFN1-xTB$^{re-par}$ (see next section). The calculations were performed in the XTB program (Bannwarth et al., 2021), version 6.4.0.

Cluster configurational sampling based on the artificial bee colony (ABC) algorithm was performed with the ABCluster program (Zhang and Dolg, 2015, 2016) using a CHARMM force field. Additional configurational sampling was performed with CREST 2.12 (Grimme, 2019; Pracht et al., 2020, 2024). All the calculations and data collection were performed with the freely available JKCS/JKQC suite of scripts (Kubečka et al., 2023a)

### 2.1.1 Cluster configurational sampling

Here we study $(SA)_n(base)_n$ clusters, with $n = 1 - 15$ and base = AM, MA, DMA and TMA. The SA–AM and SA–DMA systems were previously explored by Wu et al. (2023) and additional sampling was carried out with CREST in this work.

Configurational sampling techniques of small ($n \leq 4$) atmospheric $(SA)_n(base)_n$ clusters has previously been well-established in the literature (Temelso et al., 2018; Kubečka et al., 2019; Odbadrakh et al., 2020). However, sampling large clusters with $n \geq 5$ presents an enormous challenge. To thoroughly explore the configurational space of the clusters we apply our recently

identified configurational sampling workflow, which has been optimized towards the sampling of large cluster structures (Wu et al., 2023):

$$\text{ABC} \xrightarrow{N=10,000} \text{xTB}^{\text{OPT}} \xrightarrow{N=10,000} \text{B97-3c}^{\text{SP}} \xrightarrow[\text{filter}]{N=1,000} \text{B97-3c}^{\text{PART OPT}} \xrightarrow[\text{filter}]{N=100} \text{B97-3c}^{\text{FULL OPT}} \qquad \text{(ABC track)}$$

In brief, 10 separate configurational sampling explorations are performed with the ABCluster program ($SN = 1280$, $gen = 320$, $sc = 4$), saving 1000 lowest minima for each run. Ten parallel runs should be sufficient to model clusters consisting of

up to 15 acid-base pairs Wu et al. (2023), but might be excessive for the smallest clusters studied here. However, we kept the number of runs constant for simplicity. Ionic monomers were used in all the sampling runs. Each generated configuration is then optimized at the GFN1-xTB level of theory. A B97-3c single point energy is calculated on top of all the GFN1-xTB structures and the 1000 structures lowest in electronic energy are subsequently partially optimized with $4n$ iterations for the $(SA)_n(base)_n$ systems, at the B97-3c level. Finally, the 100 structures lowest in electronic energy based on the partial optimization are fully

optimized, and vibrational frequencies are calculated at the B97-3c level of theory.

The identified clusters were used to reparameterize GFN1-xTB using the same methodology as Knattrup et al. (2024) where the error of the electronic binding energies and gradients are minimized between GFN1-xTB and B97-3c. The three configurations with the lowest electronic energy in $(SA)_n(AM/MA/DMA/TMA)_n$ clusters ($n = 1 - 15$) resulted from the ABC track were used as the optimization set. This leads to an optimization set comprising a total of 179 clusters. The GFN1-xTB reparameterization based on this optimization set will be denoted as the GFN1-xTB$^{\text{re-par}}$. To further explore the vast configurational space of these large clusters, the identified cluster structure lowest in free energy at the B97-3c level from the ABC track is additionally used as input for sampling with CREST using the newly parameterized GFN1-xTB. Hence, we employed the following "improvement workflow" as given by Knattrup et al. (2024):

$$\text{CREST(GFN1-xTB}^{\text{re-par}}) \xrightarrow{N=100} \text{B97-3c}^{\text{FULL OPT}} \qquad \text{(CREST track)}$$

The CREST simulations were run in non-covalent interaction mode (--nci), with the GFN1-xTB$^{\text{re-par}}$ model and we employed an energy window of 30 kcal/mol (--ewin 30). We emphasize that while this overall workflow (ABC+CREST track) can very accurately identify low free energy configurations of large clusters, it is also extremely computationally demanding. However, we note that one can never be certain to locate the global minimum.

The final structures from the ABC and CREST sampling tracks are merged and unique configurations are identified based on RMSD using the ArbAlign program (Temelso et al., 2017), with an RMSD cutoff of 0.4 Å. As the largest studied system $[(SA)_{15}(TMA)_{15}]$ contains 300 atoms, the changes in hydrogen atom positions will mask the changes of the more relevant atoms by the averaging over atoms with RMSD. Hence, during the uniqueness test, we compared the geometries containing only sulfur and nitrogen atoms, and thereby local structure variations will be neglected. For instance, the rotation of a methyl group or the bending N–C–H angle does not contribute to the RMSD value.

### 2.1.2 Free energies

We calculate the binding free energies as the free energy of the cluster with respect to its individual monomers:

$$\Delta G_{\text{bind}} = G_{\text{cluster}} - \sum_i G_{\text{monomers},i} \qquad (1)$$

We also calculate the size averaged binding free energies ($\Delta G_{\text{bind}}/m$) of the clusters as the physical interpretation is the binding free energy contribution per monomer in the cluster. This quantity provides insight into the average binding properties of the cluster and offers inferred evidence about the thermochemistry associated with monomer addition. Analyzing how the average binding free energy changes with cluster size will present to us the stabilization processes occurring during cluster growth. A recent work by Sindel et al. (2022) used a similar definition to study $TiO_2$ clustering, leading to a convergence in the size-averaged binding free energies toward the formation free energy of the bulk crystal. For example, consider the difference in average binding free energy between a $(SA)_{99}(AM)_{99}$ cluster and a $(SA)_{100}(AM)_{100}$ cluster. In such large clusters, adding one extra acid-base pair results in minimal molecular rearrangement, and thus the average binding free energy remains largely unchanged. This behavior is analogous to condensation thermodynamics. In contrast, adding one $(SA)_1(AM)_1$ pair to form a $(SA)_2(AM)_2$ cluster results a huge drop in the average free energy, as the addition causes a large stabilization at such small

cluster sizes by a significant molecular rearrangement.

All thermochemistry during the sampling is performed under standard condition with temperature as 298.15 K and reference pressure as 1 atm. Enthalpy $\Delta H$ and entropy $\Delta S$ are assumed to be constant in the given temperature scale for calculation of Gibbs free energies $\Delta G$. As the default of ORCA, the quasi-harmonic approximation formulated by Grimme (2012) was applied to correct low ($< 100$ cm$^{-1}$) vibrational frequencies. All the calculated data are available in the atmospheric cluster database (ACDB) (Elm, 2019).

### 2.1.3 Convex hull approach

We previously identified that in the large SA–AM and SA–DMA cluster structures fully coordinated ions emerged, corresponding to a "solvated" ion with a solvation shell around it (Engsvang and Elm, 2022; Engsvang et al., 2023a; Wu et al., 2023). This can give a hint to when we transition from clusters, where all monomers are exposed to the exterior, to a "solvated" cluster structure more resembling the particle-phase. However, obtaining such structural information from visual inspection of 135 large clusters is difficult and prone to errors. To investigate when the first "solvation" shell appears in our cluster structures we here employed the mathematical concept of a 3-dimensional convex hull. A convex hull can be defined as the minimal convex set containing all the data, that is, it forms a polytope around the data with vertices as the "outermost" data points. Our approach is outlined as follows: The clusters are divided into monomers and these monomers are then reduced into a single 3-dimensional point at their center of mass (COM). Using the COM picture we can compute the convex hull of the monomers 140 and take this as the "solvation" shell. However, this can lead to situations where a monomer (COM) is located just inside the convex hull, i.e. it will not be interpreted as part of the convex hull even though chemical intuition would not claim it as a solvated monomer. To circumvent this issue, a rough estimate of the molecular radius is computed for all monomers. This is done simply by computing the distance from the centroid of the monomer to all atoms and averaging over the 4 largest of these distances. Then if any COM is within a distance from a convex hull face that is less than its given molecular radius, it is 145 included as the current solvation shell. After a solvation shell has been identified, the COMs included in this shell are removed from the data set and we iterate until all data points are assigned to a solvation shell. The applied algorithm is freely available at https://gitlab.com/AndreasBuchgraitz/clusteranalysis

## 3 Results and discussion

### 3.1 Evaluation of the improvement workflow

We initially evaluate how large an improvement the addition of the CREST track is compared to only employing the ABC track. Figure 1 shows the difference in free energy at the B97-3c level of theory for the lowest configuration found by the original workflow given by the ABC track and the improvement workflow given by the CREST track for the (SA)$_n$(AM/MA/DMA/TMA)$_n$ clusters, with $n = 1 - 15$. We note that the monomer count ($m$) here is simply the number of molecules in the cluster ($m = 2n$).

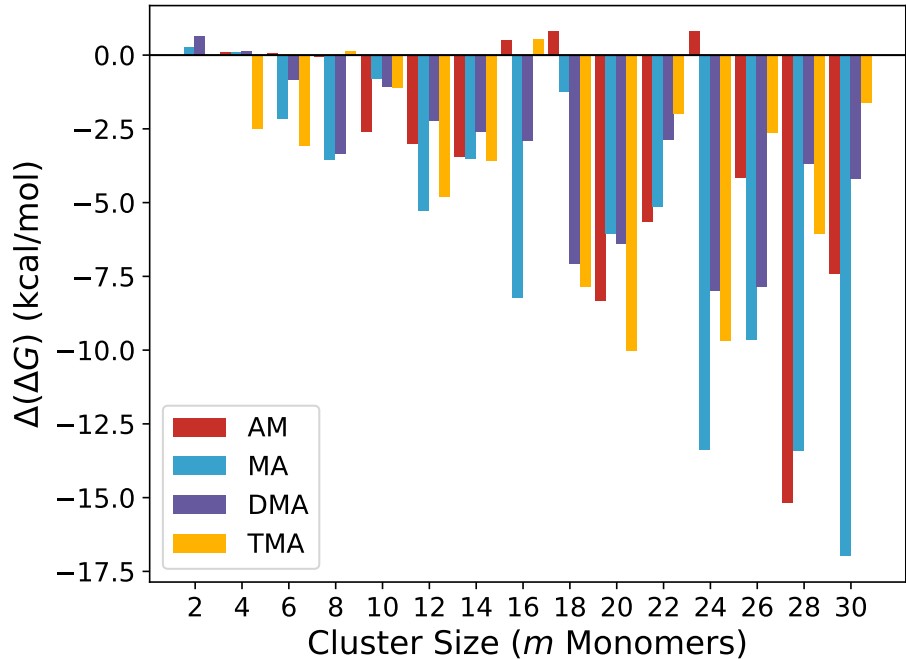

**Figure 1.** Difference in the free energy minimum found by the CREST track and the ABC track.

The addition of the CREST track significantly improves the free energy minimum found for almost all the studied clusters. The improvements scale roughly with system size where for the smallest clusters ($m = 2 - 4$) there are seen no improvement and in some cases, the located free energy minimum is slightly higher in free energy. This is caused by the potential energy surface of such small clusters being well sampled with ABCluster using rigid monomers and there is no gain by including the extra dynamic CREST step. Already for $m = 6$ the free energies after the CREST track are significantly lower. The largest improvement is seen to be up to a 17 kcal/mol lowering in the free energy for the $(SA)_{15}(MA)_{15}$ cluster. Figure 2 presents the two different $(SA)_{15}(MA)_{15}$ cluster structures, calculated at the B97-3c level of theory.

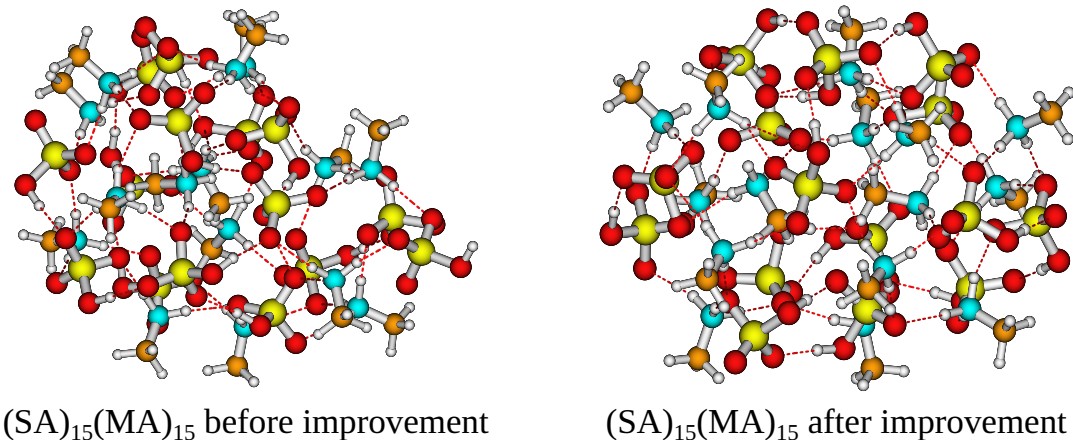

$(SA)_{15}(MA)_{15}$ before improvement          $(SA)_{15}(MA)_{15}$ after improvement

**Figure 2.** Structure of the $(SA)_{15}(MA)_{15}$ cluster lowest in free energy at the B97-3c level of theory before improvement (ABC track) and after improvement (CREST track).

It is seen that the $(SA)_{15}(MA)_{15}$ cluster after the CREST track is much more spherical compared to before. This leads to a more intricate hydrogen bond network and could explain the free energy difference. For most cluster sizes, the largest improvements are seen for the AM and MA clusters. The different hydrogen bond capacity of the base molecules most likely causes this effect. I.e. the more bulky DMA and TMA molecules will have deeper minima as there need to be a perfect match between the hydrogen bond donors and acceptors. This is in most cases relatively well captured by the ABCluster genetic algorithm. However, there are many more possible arrangements for AM and MA, which likely favor molecular dynamics sampling using CREST.

## 3.2  Cluster structures - the convex hull approach

We are interested in identifying at what cluster size we observe a "solvated" ion with a full solvation shell around it, as this could be an indication of the cluster-to-particle transition point. It should be noted, that such a solvation has previously been observed by Ling et al. (2017) in NA-AM clusters (NA = nitric acid), by DePalma et al. (2012); Engsvang and Elm (2022); Engsvang et al. (2023a) in SA–AM clusters and by DePalma et al. (2014); Wu et al. (2023) in SA–DMA clusters. Here, we further elaborate on this concept and connect it to the chemical composition of the clusters. Figure 3 presents the number of solvation shells identified with the 3-dimensional convex hull algorithm, described in section 2.1.3, as a function of the cluster size (number of monomers $m$). We used the lowest free energy clusters for each of the four systems $(SA)_n(AM)_n$, $(SA)_n(MA)_n$, $(SA)_n(DMA)_n$, and $(SA)_n(TMA)_n$ for $n = 2 - 15$ ($m = 2n$).

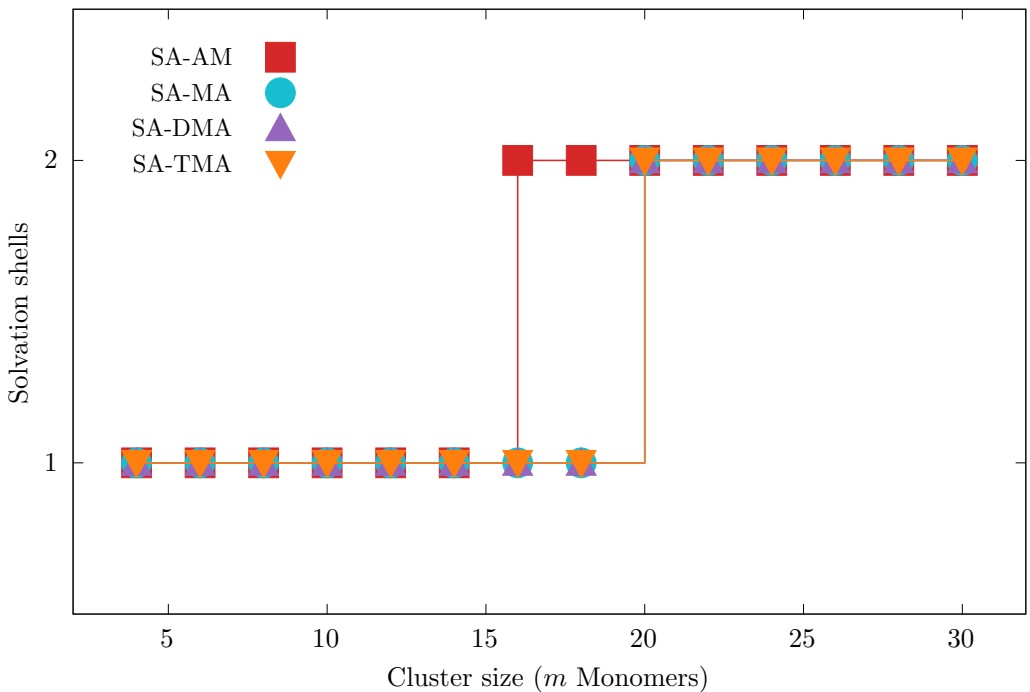

**Figure 3.** Number of identified "solvation" shells found using the convex hull approach for the lowest free energy $(SA)_n(AM)_n$, $(SA)_n$-$(MA)_n$, $(SA)_n(DMA)_n$, and $(SA)_n(TMA)_n$ systems, with $n = 2 - 15$ ($m = 2n$).

We see that for all our cluster systems we identify a maximum of 2 solvation shells at the largest sizes ($m = 30$). We find that "solvation" happens at $m = 16$ for the SA–AM system and at $m = 20$ for the SA–MA, SA–DMA, and SA–TMA systems. We find that either the bases or a bisulfate can be found in the core of the cluster depending on the cluster composition and size. In the future it would be interesting to study the emergence of the first solvation shell using semi-empirical molecular dynamics simulations.

### 3.3 Binding free energies

Figure 4 shows the calculated binding free energies of the $(SA)_n(AM/MA/DMA/TMA)_n$ clusters, with $n = 1 - 15$. The free energies are calculated at the B97-3c level of theory, at 298.15 K and 1 atm. The left panel shows the total binding free energies, the middle panel shows the binding free energies per monomer $m$ and the right panel shows the addition free energy of an acid-base pair.

As also seen in previous studies (DePalma et al., 2012, 2014; Engsvang and Elm, 2022; Engsvang et al., 2023a) the binding free energy more or less linearly decreases as a function of cluster size $m$ (see Figure 4, left panel). At $m \geq 10$, the order in the total binding free energies follow: TMA < AM < DMA < MA and no longer changes as the cluster size increases. This is an interesting trend, as SA–base cluster formation is usually connected with the gas-phase basicity of the base for small clusters.

However, for larger clusters, this appears not to be the case. This effect was already alluded to in the work by Temelso et al. (2018).

At the initial clustering (with $m = 2$), the binding strength follows the order: AM < MA < TMA < DMA. This matches observation from experiments by Jen et al. (2014) on the base stabilization of sulfuric acid dimers. The order is also relatively consistent with prior theoretical work (Kurtén et al., 2008; Olenius et al., 2017; Temelso et al., 2018; Elm, 2021a; Kubečka et al., 2023b) except the change in the order of DMA and TMA. Table 1 presents the calculated literature values for the binding free energies of the $(SA)_1(base)_1$ clusters, compared to the current work.

It should be noted that the linearly decreasing trend is dependent on the cluster composition and requires strongly binding clusters. For instance, in the previous work done by Myllys et al. (2021) (As shown in Figure 1, 6 and 10 of their paper), the $\Delta G_{bind}$ of $(SA)_1(Base)_1(H_2O)_n$ is plotted against the number of water molecules $n$ (where $n = m - 2$, with $m$ being the total monomer count, including one SA and one base molecules). In all these cases the free energy is not simply linearly decreasing as the inter–molecular interactions are quite weak. For instance, hydration of bases even gives an increasing free energy as a function of water molecules (see Myllys et al. (2021), Figure 1).

**Table 1.** Comparison between different calculated values (in kcal/mol) of the binding free energies of the $(SA)_1(base)_1$ clusters, with base = AM, MA, DMA, and TMA. All values are calculated at 298.15 K and 1 atm.

| Cluster | $a$ | $b$ | $c$ | $d$ | $e$ | $f$ |
|---|---|---|---|---|---|---|
| $(SA)_1(AM)_1$ | −6.6 | −7.3 | −7.6 | −5.6 | −8.2 | −5.9 |
| $(SA)_1(MA)_1$ | −10.0 | −10.7 | −11.5 | −7.2 | −9.8 | −8.1 |
| $(SA)_1(DMA)_1$ | −13.7 | −13.2 | −15.4 | −11.5 | −12.2 | −11.9 |
| $(SA)_1(TMA)_1$ | −15.3 | −14.3 | −15.7 | −12.6 | −11.2 | −12.8 |

[a] Kurtén et al. (2008): RI-CC2/aug-cc-pV(T+)Z//RI-MP2/aug-cc-pVDZ, harmonic

[b] Temelso et al. (2018): MP2-F12/cc-pVTZ-F12//MP2/aug-cc-pVDZ, harmonic

[c] Olenius et al. (2013, 2017): RI-CC2/aug-cc-pV(T+)Z//B3LYP/CBSB7, harmonic

[d] Elm (2021a): DLPNO-CCSD(T$_0$)/aug-cc-pVTZ//$\omega$B97X-D/6-31++G(d,p), quasi-harmonic

[e] This work: B97-3c, quasi-harmonic

[f] This work: DLPNO-CCSD(T$_0$)/aug-cc-pVTZ//B97-3c, quasi-harmonic

In most of the previous quantum chemistry studies the $(SA)_1(DMA)_1$ and $(SA)_1(TMA)_1$ clusters have very similar binding free energies. The values calculated at the RI-CC2/aug-cc-pV(T+)Z//B3LYP/CBSB7 level most likely yield too negative binding free energies (Schmitz and Elm, 2020), and the DLPNO-CCSD(T$_0$)/aug-cc-pVTZ//$\omega$B97X-D/6-31++G(d,p) level most likely underestimate the binding free energies (Myllys et al., 2016). The true value is most likely in-between these two extremes. The B97-3c calculated free energies in this work agree relatively well with the literature values and should produce the correct trends. However, the values are highly improved by refining the single point energy with a high level DLPNO-CCSD(T$_0$)/aug-

cc-pVTZ calculation. Unfortunately, such calculations are too expensive to apply to the full set of the $(SA)_n(base)_n$ clusters, with $n$ up to 15. However, this indicates that higher level single point energies should be calculated on top of the B97-3c structures to improve the values in the future. This is consistent with the conclusion by (Engsvang et al., 2023a) for large SA–AM clusters.

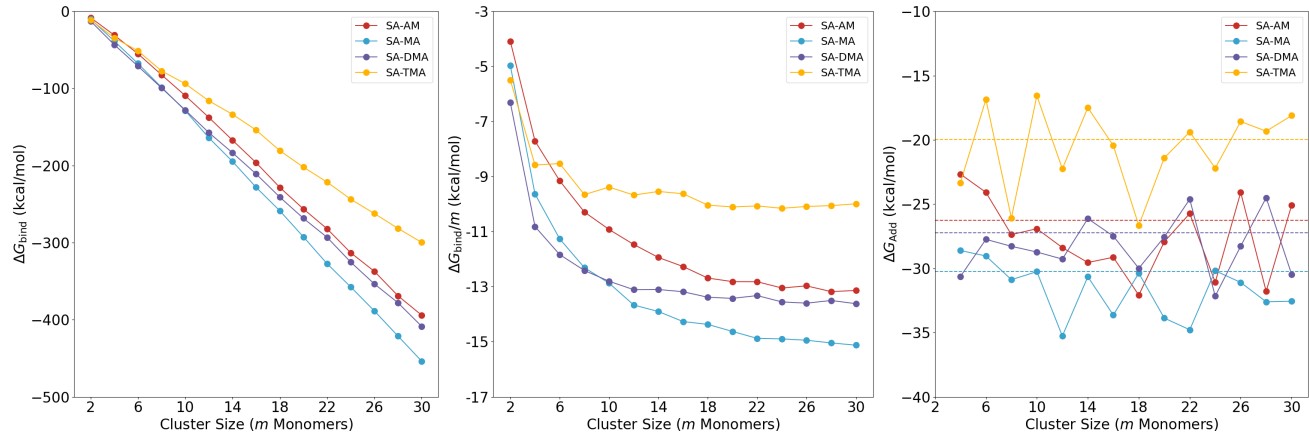

**Figure 4.** *Left:* Total binding free energies of $(SA)_n(AM/MA/DMA/TMA)_n$ clusters, $n = 1$–$15$. *Middle:* Size-averaged binding free energy contribution in the clusters. *Right:* Acid–base pair addition free energies $\Delta G_{add}$ of the four SA–Base clusters. (The dot line indicates the values of $2 \times \Delta G_{bind}/m$ with $m = 30$, the values were multiplied by two as we present pair wise addition here). Data of SA-AM/DMA are from our prior work (Wu et al., 2023), with additional sampling carried out in this work.

In the middle panel of Figure 4 it can be seen that at $m \geq 20$, the curve of $\Delta G_{bind}/m$ levels out and the gradient of size-averaged binding free energy $\Delta G_{bind}/m$ approaches to zero. At this point the binding strength order of TMA < AM < DMA < MA also stays consistent. The SA–MA system exhibits the highest stability with the most negative binding free energy across the $m \geq 10$ range converging at a value of $-14.9$ kcal/mol, at $m = 30$. This is closely followed by the SA–DMA system, which

converges at a value of $-13.4$ kcal/mol, at $m = 30$. The SA–AM system is slightly less stable ($-13.1$ kcal/mol, at $m = 30$), while the SA–TMA system has the highest binding free energy, indicating it is the least stable of the four systems modeled ($-10.1$ kcal mol$^{-1}$, at $m = 30$). Also, within $m \leq 8$ the stability of the SA–TMA system is highly dependent on the total number of SA-TMA pairs. For $m = 4$, our calculations show that the $(SA)_2(TMA)_2$ is less stable than $(SA)_2(DMA)_2$ which is in agreement with Elm (2021a) and Kubečka et al. (2023b). The low stability of the SA–TMA clusters at large sizes can be

understood by the high steric hindrance introduced by the three methyl groups in TMA. In a similar fashion, the change in the order of MA becoming more stable than DMA at $m = 10$ can be attributed to a combination of binding strength, hydrogen bond capacity, and steric hindrance.

As shown in the right panel of Figure 4, the acid–base pair addition free energies $\Delta G_{add}$ fluctuate around the convergence value of the size-averaged binding free energy per monomer, $\Delta G_{bind}/m$ (indicated by the dotted lines). Since the addition free

energy $\Delta G_{add}$ is the accurate quantity for estimating cluster stability, the relation between addition free energy $\Delta G_{add}$ and size–averaged binding free energy $\Delta G_{bind}/m$ is significant. Mathematically, taking the endpoint at $m$ of the size-averaged

binding free energy $\Delta G_{\text{bind}}/m$ corresponds to calculating the average of the addition free energies up to cluster size $m$. This approach helps determine the cluster size $m$ at which fluctuations in $\Delta G_{\text{bind}}$ become negligible relative to the entire system. Furthermore, the convergence value of $\Delta G_{\text{bind}}/m$ can be used to estimate the free energies of acid–base pair additions to large

clusters.

At large $m$, the low gradient of size-averaged binding free energies implies that at this point the average addition of monomers to the cluster does not change the uptake properties. We interpret this as the cluster showing particle-like properties. This also illustrates that when studying large clusters in the future we do not need to go beyond 20 monomers as the properties are already well converged at this point. On the other hand, there is a large stabilization achieved by adding more

acid–base pairs for $m \leq 8$. Hence, we can divide the cluster formation process into two different regimes: An initial *"cluster stabilization"* regime for $m \leq 8$ and a *"freshly nucleated particle (FNP)"* regime at $m \geq 20$. The *"cluster stabilization"* regime is highly dependent on the identity of the base molecule and governs the initial particle formation rate, while the *"freshly nucleated particle"* regime governs the particle growth. We will denote the transition between these two regimes as the cluster-to-particle transition regime. The observed leveling out in the average free energies also coincides with the emergence of the

second solvation shell observed in 3.2. Hence, we suggest defining the cluster-to-particle transition point around $m = 16 - 20$ and will denote clusters that have passed this point as "freshly nucleated particles (FNPs)". We note that the cluster-to-particle transition point presented here is conceptually new and is fundamentally different from the concept of a "critical cluster" in classical nucleation theory, which resembles a maximum on the free energy surface. It should be mentioned that the onset of FNPs is highly dependent on the cluster composition, temperature, and concentration of the clustering monomers. It should be

further noted that we only study the FNPs in increments of acid-base pairs in the current work and that the cluster-to-particle transition point might change once data on monomer additions/evaporation becomes available.

### 3.4    Free energies at given conditions

Using the free energies calculated above, it is possible to calculate the binding free energies under specific conditions of monomer concentrations and temperature. This will indicate whether these FNPs will actually be formed at realistic atmo-

spheric conditions. The self-consistent distribution function proposed by Wilemski and Wyslouzil (1995) was employed to establish the monomer free energies as zero. At given conditions the "actual" binding free energies are calculated by Halonen (2022): $\Delta G_{\text{bind}}(\boldsymbol{p}) = \Delta G_{\text{bind}}^{\circ} + RT(1 - \frac{1}{n}) \sum_i \ln(\frac{p_i}{p_{\text{ref}}})$. where $p_{\text{ref}}$ corresponds to a reference pressure (1 atm) and $p_i$ represents precursor (which in our discussion is monomer) partial pressures. This equation differs from previous work on cluster formation at "actual" conditions as these incorrectly generalized the unimolecular nucleation equation. Thus, the equa-

tion satisfies self-consistency also for multi–component systems, i.e., having zero free energies for all precursors. We testing the different formulations of the actual free energies at given conditions and found no deviations between the calculated free energies in the current work.

Figure 5 shows the actual binding free energies $\Delta G_{\text{bind}}$ of the $(SA)_n(AM/MA/DMA/TMA)_n$ clusters ($n = 1 - 15$), under given conditions. The figures are plotted as a function of the number of monomers ($m$) in the clusters ($m = 2n$). We studied

two temperatures of 298.15 K and 278.15 K, given by the red and blue shadings, respectively. The concentration of SA was

fixed at [SA] = $10^6$ molecules cm$^{-3}$. We studied two different concentrations of the bases. A "High Conc." limit with [AM] = 10 ppb, [MA] = [DMA] = [TMA] = 10 ppt and a "Low Conc." limit with [AM] = 10 ppt, [MA] = [DMA] = [TMA] = 1 ppt. We note that the chosen values of the amines should be viewed as an upper limit in non-polluted environments. Similarly, the concentration of SA could also easily exceed $10^6$ molecules cm$^{-3}$ in many environments.

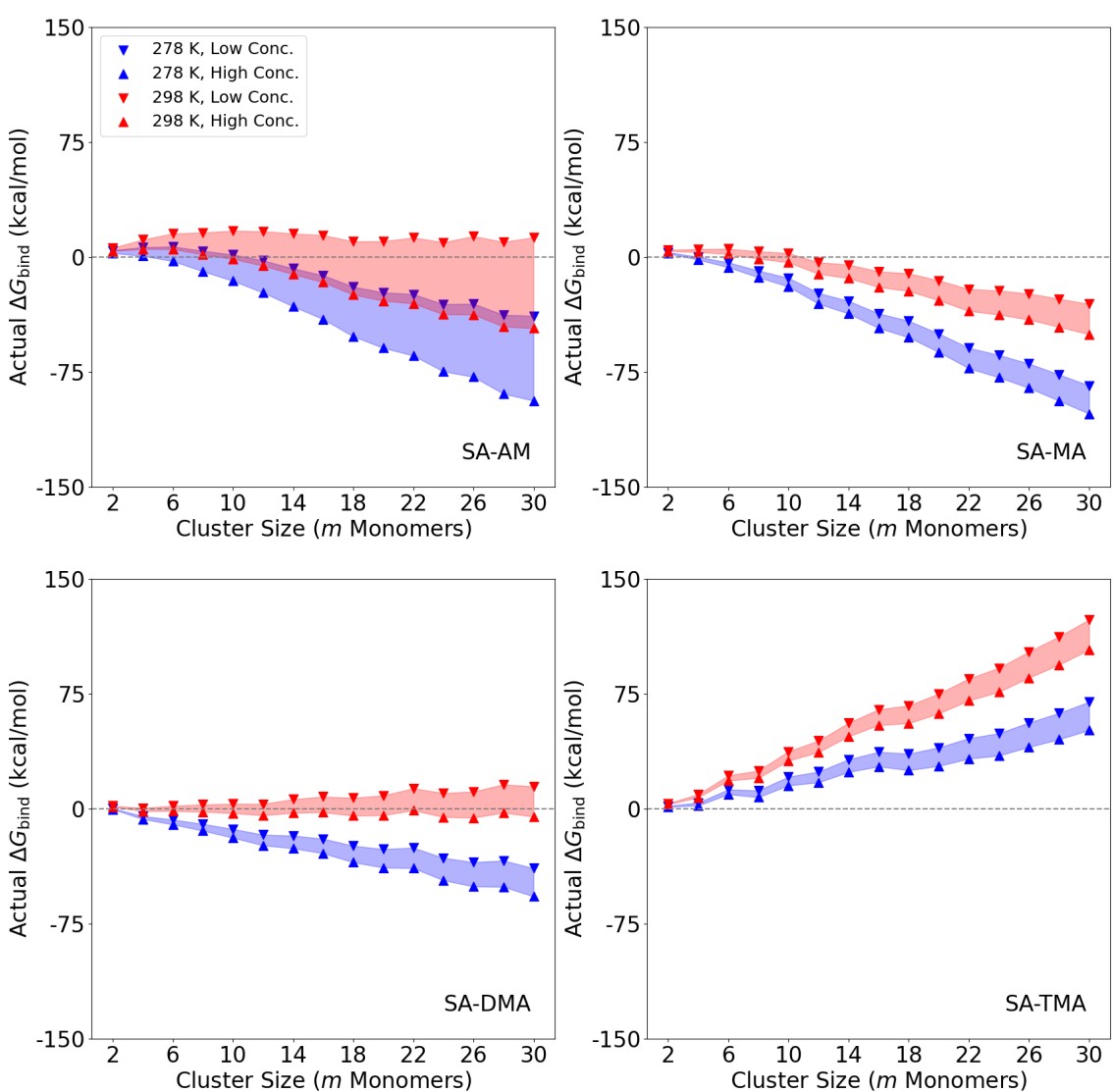

**Figure 5.** Binding free energies $\Delta G_{\text{bind}}$ of the $(SA)_n(AM/MA/DMA/TMA)_n$ clusters ($n = 1$–15, $m = 2n$) at given conditions of temperature and concentration. High temperature (298.15 K) where the concentration range is filled with red shading, and accordingly low temperature (278.15 K) by blue shading. [SA] was fixed at $10^6$ molecules cm$^{-3}$. "High Conc." refers to high concentration, with [AM] = 10 ppb, [MA] = [DMA] = [TMA] = 10 ppt (Up-pointing triangle). "Low Conc." refers to low concentration with with [AM] = 10 ppt, [MA] = [DMA] = [TMA] = 1 ppt (Down-pointing triangle).

It is seen that temperature plays a more important role than the monomer concentrations for all systems. Increasing the concentration of the base by one order of magnitude, the free energy decreases by $\sim 2.5$ kcal/mol per monomer. While decreasing the temperature from 298.15 K to 278.15 K the free energy decreases by $\sim 6$ kcal/mol per monomer. For SA–AM there is a nucleation barrier at both temperatures, which should limit the cluster formation process under such conditions. This is consistent with the previous work by Olenius et al. (2013) and Besel et al. (2019), which showed that ions are important for boosting SA–AM nucleation. It should be mentioned that the "actual" nucleation barrier of the system can not be determined based solely on the 1:1 acid-to-base ratio composition. Monomer condensation could potentially lead to a change in the barrier. However, in most SA–base systems the 1:1 ratio is the most stable composition (Olenius et al., 2013; Elm et al., 2017a) and if the 1:1 ratio does indeed show a barrier, then the $\pm 1$ acid/base molecule systems will have to cross that as well.

The SA–AM system is highly dependent on concentration, which is caused by the larger concentration range (10 ppt–10 ppb) compared to MA, DMA, and TMA (1–10 ppt). The SA–MA system also has a nucleation barrier at 298.15 K and low concentration (1 ppt), but no barrier at the other studied conditions. The SA–DMA cluster system is seen to form without a nucleation barrier at a low temperature (278.15 K). This finding is consistent with the previous work by Olenius et al. (2013) and the experiments at CLOUD, which showed that SA–DMA cluster formation occurs at the kinetic limit at temperatures of 278.15 K or below (Kürten et al., 2014; Kürten et al., 2018).

Interestingly, in all cases of $m \geq 6$, the SA–TMA clusters are very unstable and hardly bind with SA under any conditions. This indicates that TMA is only important in the *"cluster stabilization"* regime and does not help grow the particles at larger sizes. This finding is consistent with previous quantum chemical studies (Kubečka et al., 2023b) and observations (Cai et al., 2023). This makes sense from a molecular perspective, as TMA, after proton transfer from SA, only has one hydrogen bond donor and the three bulky methyl groups will lead to high steric hindrance. Hence, the hydrogen bond capacity of the base is quite important for the cluster growth. However, this contradicts the experimental results of Glasoe et al. (2015), where 1.8 nm sulfuric acid–base particle formation followed an AM < MA < DMA < TMA trend. As Glasoe et al. (2015) used quite high concentration of sulfuric acid ([SA] = $10^9$-$10^{10}$ molecules cm$^{-3}$), one reason for this discrepancy could be our fixed 1:1 ratio of sulfuric acid to bases. Hence, in the future, it might be worth investigating large clusters where the SA:base ratio is higher, instead of the usual 1:1 ratio.

The fact that the bases behave very differently as a function of the number of monomers in the cluster could indicate that SA–mixed-base systems are worth studying in the future. For instance, it is very likely that strong bases such as DMA and TMA are primarily important in the very initial steps and that the subsequent growth is entirely driven by the weaker bases AM and MA. This has previously been hypothesized for smaller clusters (Elm, 2017; Elm et al., 2017b; Elm, 2021b), but not definitively proven. Hence, large SA–AM–DMA or SA–MA–DMA clusters up to the cluster-to-particle transition point will

be worth studying in the future. This will also allow us to study base substitution in such systems and investigate whether ammonia is efficiently substituted by DMA as illustrated by Kupiainen et al. (2012) for smaller clusters.

## 3.5  Cluster populations

A common assumption when studying atmospheric molecular clusters is to only use the lowest free energy configuration for describing the cluster properties. Hence, it is assumed that the lowest free energy configuration is dominantly populated and that the global minimum is reached rapidly via molecular re-arrangement. However, whether this is valid for larger clusters remains unknown. To investigate this aspect we calculated the populations of the four lowest free energy configurations for the studied SA–base systems. Different cluster configurations were determined via RMSD as described in section 2.1.1. As a method to tell conformational differences, RMSD is less intuitive than measures such as internal coordinates (bond lengths, angles, etc.) when applied to small molecules or clusters. For large systems such as proteins, supramolecules, or large clusters, all-atoms RMSD is computationally expensive and has the risk that the geometrical information of interest is being averaged out by the large number of atoms, leading to a low signal-noise ratio. In our cases of studying large clusters, the detailed difference of the monomer side group (e.g. rotation of methyl or hydroxy groups) should not, aside from minor inductive effects, affect the intermolecular interactions significantly. Similarly, in protein science, a widely-used simplification is to use $\alpha$-C atoms to represent amino acids, and RMSD or other distance-based metrics are then afterward calculated based on only the positions of $\alpha$-C atoms (Lazar et al., 2020). Hence, during the comparison, we compared the geometries containing only sulfur and nitrogen atoms. Thereby in this comparison, the RMSD contributed to the difference between monomer sidechains are neglected.

Figure 6 shows the configurational population of the $(SA)_n(Base)_n$ systems, with $n = 1 - 15$, plotted as a function of the monomer count ($m = 2n$). Only the four lowest free energy configurations were plotted, calculated at 298.15 K and 1 atm. This is also the reason for the populations not summing up to 100% in all cases.

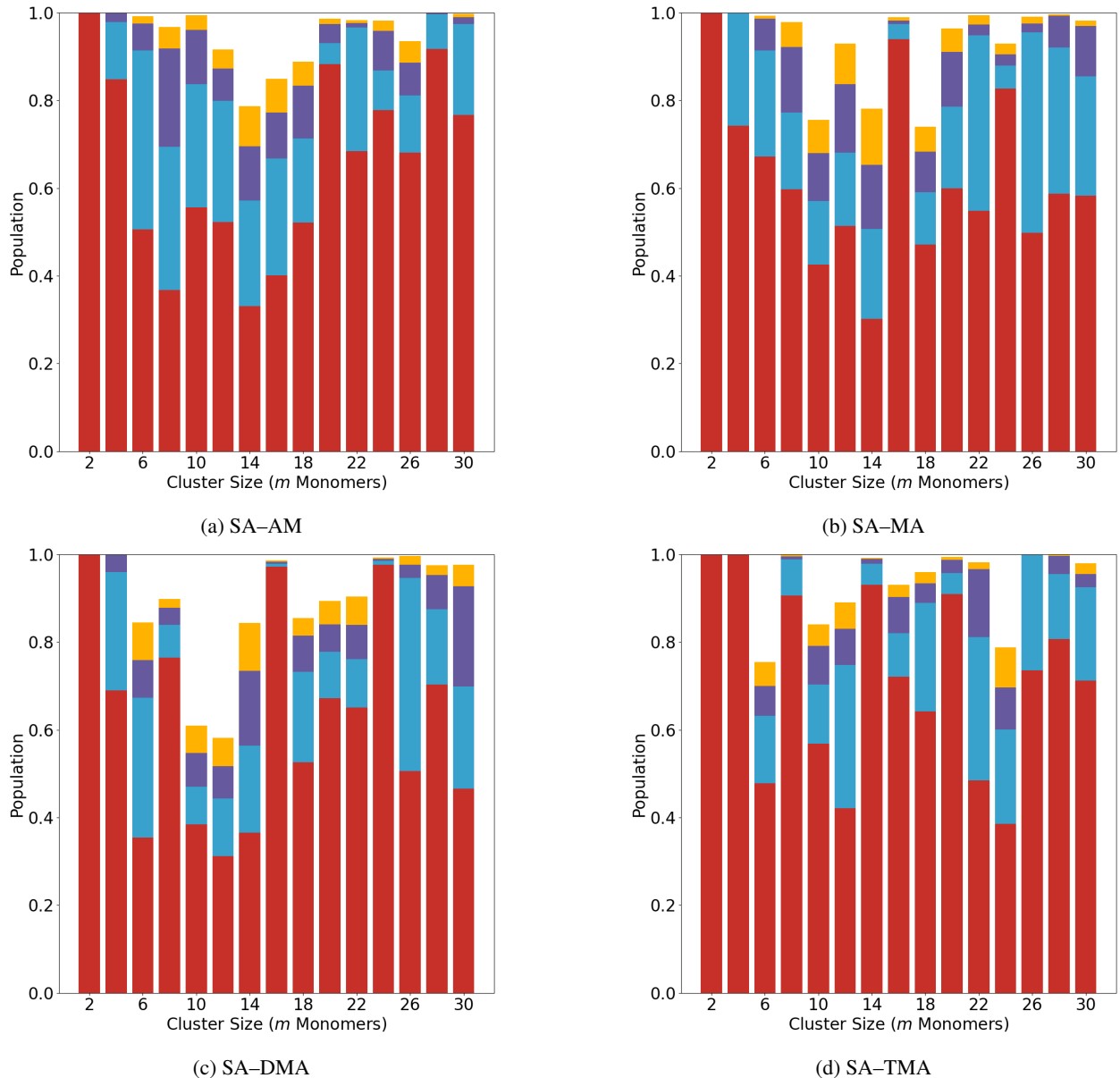

**Figure 6.** Population distribution of lowest four configurations of the $(SA)_n(AM/MA/DMA/TMA)_n$ clusters ($n = 1$–$15$, $m = 2n$) clusters.

From Figure 6 it is seen that the lowest configuration is fully populated at $m = 2$. In general, a decreasing trend of the population weight of the lowest free energy configuration is seen from $m = 4-14$, indicating that as the cluster size grows, more configurations will co-exist. At sizes larger than $m = 14$, the population of the lowest configuration begins to increase again. This is counter-intuitive and is likely an artefact of the averaging nature of the RMSD calculation. Calculating RMSD with only S and N atoms removes the contribution from the methyl group rotation or the bending of the S-O-H angle. However, for

large clusters such as those studied here, the geometric difference contributed by certain local substructure in the cluster is lost when averaging over the total number of atoms. We note that in order to properly identify unique configurations methods other than conventional RMSD should be tested. In the future we will test other methods for isolating the different configurations such as mass-weighted RMSD and contact maps.

The binding free energies of the lowest four configurations are listed in the supporting information (See table S3 and table S4). The energy gap between lowest the second lowest configurations is in most cases below 1 kcal/mol (largest value of 2.9 kcal/mol appears for $(SA)_8(DMA)_8$). It should be noted that configuration above 3 kcal/mol from the lowest free energy minimum will have a negligible contribution to the multi-configurational free energy at 298 K (Partanen et al., 2016). Nevertheless, in all cases we can see a large weight ($\geq 30\%$) on the lowest free energy cluster structure. Furthermore, by comparing the binding free energies of the lowest-energy configurations in Table S1 with the multi-conformer binding free energies in Table S2, it is evident that including all identified configurations (filtered by RMSD with threshold of 0.4 Å) reduces the effective binding free energies by less than 1 kcal/mol. This indicates that when looking at the properties of these large cluster systems one can safely study the 1–2 clusters lowest in free energy without introducing large uncertainties. Hence, in the future we can study how the FNPs take up vapour molecules, how evaporation occur from them and how chemical reactions occur at the surface by using the lowest clusters found here.

## 4   Conclusions

In this work we presented quantum chemical modeling of large $(SA)_n(AM/MA/DMA/TMA)_n$ clusters, with $n = 1-15$ (cluster size $m = 2-30$), at the B97-3c level of theory. A comprehensive configurational sampling protocol was applied to locate the lowest free energy cluster structures. When there are around $16-20$ monomers ($m$) in the cluster, we see the emergence of the first solvation shell, where an ion is fully coordinated inside the cluster core. The binding free energies (at 298.15 K and 1 atm) per monomer in the cluster, showed that the cluster growth process can be divided into a *"cluster stabilization"* regime for $m \leq 8$ and a *"freshly nucleated particle (FNP)"* regime at $m \geq 20$, where the the structure and stability of the molecular cluster more resemble particle-like properties. Consistent with previous studies we find that the *"cluster stabilization"* regime is highly dependent on the clustering base molecule. Based on these findings we define the cluster-to-particle transition point as the onset of FNPs to be around 16–20 monomers.

Studying the free energies at given conditions we find that the SA–AM and SA–MA systems have a nucleation barrier, SA–DMA form clusters below 278.15 K without a barrier, and SA–TMA does not form stable clusters at $m \geq 6$ monomers. This contradicts experimental results on 1.8 nm particles and could indicate that the 1:1 ratio of acid-to-bases is not the most likely growth pathway for all of the larger clusters. Hence, under realistic experiment conditions, the presence of water and other species might enable the growth of SA–TMA clusters.

We studied the cluster populations and found that a high weight is placed on the lowest free energy cluster structure. This indicates that the lowest free energy cluster configurations can be used to study the properties of these clusters in the future.

The fact that the different bases appear to be very important in the different regimes could indicate that large SA–mixed-base clusters should be studied in the future, to disentangle which bases are important for nucleation and which are important for the growth. In addition, the growth of the clusters should also be further investigated, i.e. calculating clusters that differ from the 1:1 acid-to-base ratio. This will also enable the possibility of performing cluster dynamics simulations. Finally, the inclusion of water should also be investigated, as it might have a large influence on the cluster stabilities and in extension the cluster-to-particle transition point.

*Code availability.* The code used to compute the solvation shells using the convex hull algorithm is available at: https://gitlab.com/AndreasBuchgraitz/clusteranalysis

*Data availability.* All the calculated structures and thermochemistry are available in the Atmospheric Cluster Database (ACDB) (Elm, 2019)

*Author contributions.* Conceptualization: J.E.;
Methodology: H.W., Y.K., A.B.J., J.E.;
Software: A.B.J.;
Formal analysis: H.W., Y.K., A.B.J.;
Investigation: H.W., Y.K., A.B.J.;
Resources: J.E.;
Writing - original draft: H.W., Y.K., A.B.J., J.E.;
Writing - review & editing: H.W., Y.K., A.B.J., J.E.;
Visualization: H.W., Y.K., A.B.J.;
Project administration: J.E.;
Funding acquisition: J.E;
Supervision: J.E.

*Competing interests.* At least one of the (co-)authors is a member of the editorial board of Aerosol Research. The authors have no other competing interests to declare.

*Acknowledgements.* Funded by the European Union (ERC, ExploreFNP, project 101040353). Views and opinions expressed are however those of the authors only and do not necessarily reflect those of the European Union or the European Research Council Executive Agency. Neither the European Union nor the granting authority can be held responsible for them.

The authors thank the Independent Research Fund Denmark grant number 9064-00001B for financial support. This work was funded by the Danish National Research Foundation (DNRF172) through the Center of Excellence for Chemistry of Clouds.

The numerical results presented in this work were obtained at the Centre for Scientific Computing, Aarhus https://phys.au.dk/forskning/faciliteter/cscaa/.

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
