# Peer review of "Cluster-to-particle transition in atmospheric nanoclusters"

_Aerosol Research, 2024_

## Author Comment (AC1)

We would like to thank the reviewers and Shuai Jiang for the constructive comments. In the revised version of the manuscript we have carefully taken each comment into account and believe it has improved the study. The reviewers' and the community comments have been reproduced in blue text below, followed by our point-by-point replies.

**Community Comment by Shuai Jiang**

This study investigates the structures and thermodynamics of larger acid-base clusters, providing important insights into freshly nucleated particles based on structural features and thermodynamics. It offers valuable understanding of the phase transition characteristics of aerosol nucleation molecular clusters and merits publication. The comments are listed as follows:

1. There should be evidence or references to support the statement that "we identified large uncertainties in the calculated thermochemistry, which was attributed to insufficient configurational sampling."

> **Authors' reply:** We agree with the reviewer that the statement should be further justified. The referral to the large error in the thermochemistry is to the work done by Engsvang cited in the sentences above. We have added the references to the sentence on page 2, line 55 to clarify that we are referring to that work here as well.

2. Regarding parallel sampling, why is the number of parallel samples always 10 for different kinds and sizes of clusters? Should it not be larger for larger clusters?

> **Authors' reply:** Indeed, larger clusters are significantly more difficult to sample correctly. In our previous work, [1] we found that 10 parallel runs were necessary to adequately sample clusters consisting of up to 15 acid-base pairs. Hence, the 10 runs employed in the current work should be sufficient for sampling the largest 10 acid-base pair clusters. However, it might be overkill for the smaller clusters. It should of course be mentioned that no matter how many configurational sampling runs performed, we can still miss a certain local minima. To further clarify this aspect we have added the following sentence:
>
> **Added sentence, Page 3, line 84:**
> Ten parallel runs should be sufficient to model clusters consisting of up to 15 acid-base pairs [1], but might be excessive for the smallest clusters studied here. However, we kept the number of runs constant for simplicity.

3. Configurational sampling on the PES is an NP-hard problem, making it extremely challenging to find the global minimum for large clusters. While parallel sampling is better than single sampling, how can you prove that the true global minimum is found for large clusters, or how confident are you in the sampling of large clusters?

> **Authors' reply:** We agree with Shuai Jiang that configurational sampling is extremely challenging, especially for large clusters. In reality, one can newer be certain that the "true" global minimum is found, all we can hope for is finding something that is as close as possible. In our previous work on large clusters [1, 2] we have carefully assessed how sampling must be improved for larger clusters. Our current methodology is quite extensive, both employing many ABCluter runs and metadynamics simulations via CREST. Hence, with our current methodology, we are quite confident that we are finding a minimum fairly close to the global minimum. To further comment on this aspect, we have modified the following sentence:
>
> **Modified sentence, Page 4, line 101:**
>
> **From:**
> We emphasize that while this overall workflow (ABC+CREST track) can very accurately identify the lowest free energy cluster structure of large clusters, it is also extremely computationally demanding.
>
> **To:**
> We emphasize that while this overall workflow (ABC+CREST track) can very accurately identify low free energy configurations of large clusters, it is also extremely computationally demanding. However, we note that one can never be certain to locate the global minimum.

4. In Figure 4, it is interesting to note that SA-MA is more stable than SA-DMA. I suggest adding more discussions about the main driving force behind this change in order.

**Authors' reply:** We agree with Shuai Jiang that it is interesting that the cluster stability does not follow the basicity of the base as usually seen for small clusters. We believe this trend is caused by a combination of the following trends of the base: 1) the basicity, 2) the hydrogen bond capacity (i.e. the number of hydrogen bond donors) and 3) the steric hindrance of the methyl groups in the base. For these clusters MA reaches a sweet-spot where these factors are more favourable for larger clusters compared to DMA. However, as also shown in Figure 4, DMA clusters are still more stable in the initial *"cluster stabilization"* regime. We believe this aspect is already covered in the following sentence in the manuscript on page 9, line 212:

*"The low stability of the SA–TMA clusters at large sizes can be understood by the high steric hindrance introduced by the three methyl groups in TMA. In a similar fashion, the change in the order of MA becoming more stable than DMA at $m = 10$ can be attributed to a combination of binding strength, hydrogen bond capacity, and steric hindrance."*

5. The cluster-to-particle transition is very intriguing for those interested in the physical chemistry of atmospheric aerosol nucleation. Additionally, further discussions on the implications for ambient atmospheric aerosol formation should be included to broaden its impact. For example, a more detailed discussion on how the findings of this study could help reduce uncertainties in climate predictions would significantly enhance the manuscript's contribution to the field.

**Authors' reply:** We completely agree that this is conceptually new, and that the results will guide the definition on when we can classify an assembly of molecules as a cluster or as a particle. This will allow us to better disentangle nucleation and growth in aerosol process model. However, at the current stage to directly extrapolate these findings to the implication of reducing the uncertainties in climate predictions would be speculative at best. Hence, we will refrain from commenting on this aspect in the current contribution, but will keep it in mind in future work.

**Anonymous Referee 1**

Wu et al. present a definition of freshly nucleated particles based the determination of thermodynamic properties of large sulfuric acid-base clusters (where base = ammonia/methylamine/dimethylamine/trimethylamine). Their study allows to clearly distinguish the boundary between discrete cluster configurations and bulk particles. This study is of fundamental importance in understanding the cluster-to-particle transition process in sulfuric-based clusters. The manuscript is very well written and results are of substantial relevance to Aerosol science. Therefore, I recommend publication to Aerosol Research after the following minor comments have been addressed.

1. Introduction. It would be better to give a reference to the recent IPCC report.

**Authors' reply:** We agree with the reviewer that the most recent report should be used. Hence, the IPCC report has been changed to the 2023 version on page 1, line 18.

2. Subsection 2.1.1. For each cluster type, the authors start from thousands randomly generated configurations, ending up with one assumed cluster structure lowest in free energy that is used in further thermochemical analysis and modeling. It is obvious that real life situations would involve multiple configurations. Although they report in Subsection 3.5 that two lowest energy configurations are enough when determining the clusters properties, the populations of two lowest energy configurations can rarely exceed 60% in some cases. Can't the thermal averaging of multiple conformers be considered in this case?

**Authors' reply:** Indeed, multiple configurations are most likely co-existing. However, for these large clusters we ran into difficulties in how to efficiently and unambiguously define different "configurations" as also explained in section 2.1.1 and 3.5. The issue is that due to the large number of atoms in the large clusters RMSD is not sensitive enough to differ between configurations as we average over all atomic positions. A temporary fix in the current study was to calculate the RMSD using only the sulfur and nitrogen atoms. This is of course not a satisfactory result if we are to calculate the multi-conformer free energies. We are currently looking into other means of isolating

the different configurations such as mass-weighted RMSD and contact maps. In the future we will also look into molecular dynamics of the clusters and in that regard, multiple configurations and the interchange between these will be taken into account. To comment on this aspect, we have added the following sentence to the manuscript:

**Added sentence, Page 15, line 311:**
We note that in order to properly identify unique configurations methods other than conventional RMSD should be tested. In the future we will test other methods for isolating the different configurations such as mass-weighted RMSD and contact maps.

3. Subsection 2.1.2. The authors used the convex hull approach to investigate when the first solvation shell appears. Don't they think that molecular dynamics simulations could be a better approach since they provide the "real" dynamics of clusters evolution with time?

**Authors' reply:** Indeed, the emergence of the first solvation shell in this aspect is purely based on a "static" thermodynamic picture. Studying the first solvation with molecular dynamics is very interesting, but also not trivial. To directly see the cluster-to-particle transition point would require collision trajectory simulations, where molecules (either SA or base) are colliding with the smaller cluster leading to the composition at the cluster-to-particle transition point. Prior to the present work it would not be possible to know at what point this would happen. In addition, this would most likely not be possible to study with classical force fields, as bond breaking is not described. We are currently developing collision simulations using semi-empirical MDs and hopefully in the future this will be possible to study. However, at the moment we will have to leave this as a teaser as much more work is needed. To reflect on this aspect, we have added the following sentence to the manuscript:

**Added sentence, Page 7, line 167:**
In the future it would be interesting to study the emergence of the first solvation shell using semi-empirical molecular dynamics simulations.

4. Subsection 3.5. What is the energy difference range between the global minimum and other three lowest free energy configurations?

**Authors' reply:** We agree with the reviewer that it would be beneficial to comment on the free energy range between the lowest minima and the higher lying minima. To address this issue, we have added the following the sentence below to the manuscript. In addition, we have added the raw data to the supporting information.

**Page 15, Line 314:**
The binding free energies of the lowest four configurations are listed in the supporting information (See table S3 and table S4. The energy gap between lowest the second lowest configurations is in most cases below 1 kcal/mol (largest value of 2.9 kcal/mol appears for $(SA)_8(DMA)_8$). It should be noted that configuration above 3 kcal/mol from the lowest free energy minimum will have a negligible contribution to the multi-conformer free energy at 298 K.

5. Page 15, Line 312: "...to disentangle which bases that are important for nucleation and which that are important for the growth." Remove "that" at the two places.

**Authors' reply:** We agree that the word "that" should be removed in these intances. The following changes have been made:

**Modified sentence, Page 15, line 343:**
"...to disentangle which bases  are important for nucleation and which  are important for the growth."

6. The authors should choose to use either the unit kcal mol-1 or kcal/mol both in the text and in the figures/tables and not use both in a sporadic way.

**Authors' reply:** We completely agree with the reviewer that the units should be consistent throughout the manuscript. All $kcal \cdot mol^{-1}$ have been rewritten to kcal/mol to be consistent with the figures.

7. Check the good spelling of the author named "Sipilä" used at line 25 and in the reference list.

> **Authors' reply:** We apologize for the misspelling of the reference. This has been corrected in the revised manuscript.
* * *
**Anonymous Referee 2**

The study by Wu et al. addresses the question of molecular-cluster-to-particle transition in aerosol nucleation. Resolving such transition sizes is important for being able to understand and represent secondary particle formation and growth in atmospheric models. Assessing transition processes has mostly been impossible due to the lack of reliable theoretical and computational approaches for describing the chemistry of growing molecular complexes of >1nm, and therefore this work makes an important contribution.

The study is relevant for the journal scope, and the methods and results are generally sound. I have a few comments that I ask the authors to address before I can recommend publication.

1. It needs to be clarified that the suggested criteria for a cluster becoming a particle address the structure and stability of the molecular complex, and not the actual chemical or thermodynamic properties as compared to liquid properties, as described e.g. by well-established ionic liquid models such as E-AIM and AIOMFAC.

Therefore, for clarity, I suggest to reformulate statements such as "...freshly nucleated particle (FNP) regime where the properties more resemble bulk particle-like properties", as this is easily (mis)understood as bulk thermodynamic properties, e.g. protonation states, activity coefficients and Kelvin effect.

> **Authors' reply:** We agree with the reviewer that it should be clarified that the proposed criteria is referring to the molecular cluster and not bulk thermochemistry such as described by the s E-AIM or AIOMFAC models. To clarify, we have removed the word "bulk" throughout the manuscript.
>
> **Modified sentence, Page 1, line 7:**
> that act as a boundary between discrete cluster configurations and →large particles.
>
> **Modified sentence, Page 4, line 119:**
> This can give a hint to when we transition from clusters, where all monomers are exposed to the exterior, to a "solvated" cluster structure more resembling the  particle-phase.
>
> **Modified sentence, Page 9, line 217:**
> We interpret this as the cluster showing  particle-like properties.
>
> **Modified sentence, Page 15, line 331:**
> ...where the →structure and stability of the molecular cluster more resemble  particle-like properties.

2. I'm not sure if I understand the meaning of the quantity that is used to define "FNP" in Figure 4, left panel. By looking at the values in Figure 4 and as stated in the text, this quantity seems to be the binding free energy divided by the number of molecules in the complex, i.e. DeltaG/m. As a result, the shape of the graphs mainly illustrates the reciprocal of the monomer number, i.e. $1/m$, $m = 2, 4, 6, \ldots$

Especially the definition in the Abstract is strange: "we define the onset of FNPs as (...) when the gradient of the change in free energy per monomer (m) approaches zero" since DeltaG/m is not a gradient nor does it approach zero in Figure 4.

I recommend to use instead the change in DeltaG upon addition of monomer pair, i.e. the actual gradient Delta(DeltaG); maybe this was also the original idea of the authors based on what is written in the Abstract? Intuitively, the discrete "cluster-like" effects should gradually disappear with increasing molecular complex size as the effects of the positions, orientations and bonds of individual molecules become less important. The transition to a more "particle-like" phase can indeed be expected to be manifested by the gradient of the size-dependent thermochemical properties exhibiting a smoother behavior with increasing size, instead of larger and irregular changes.

> **Authors' reply:** Indeed, we meant the actual gradient in the form of $\Delta(\Delta G)$. However, we agree

that the terminology might have been too loosely formulated. The relevant content in abstract and section 3.3 has been rephrase to prevent potential misunderstanding.

**Page 1, Line 8:**
We define the onset of FNPs as when one or more ions are fully solvated inside the cluster and when the gradient of the →size-averaged binding free energy approaches zero.

**Page 9, Line 204:**
In the right panel of Figure 4 it can be seen that at $m \geq 20$, the curve of $\Delta G_{\text{bind}}/m$ levels out and →the gradient of size-averaged binding free energy $\Delta G_{\text{bind}}/m$ approaches to zero.

**Page 9, Line 216:**
→At large $m$, the low gradient of size-averaged binding free energies implies that at this point the average addition of monomers to the cluster does not change the uptake properties.

3. (i) Even when the properties start to resemble those of liquid particles as the cluster size increases, the stability is still expected to exhibit a size dependence at nanometer sizes through the surface curvature effect, i.e. the Kelvin effect (e.g. Factorovich et al., Am. Chem. Soc. 136, 4508-4514, 2014). Can you comment on if and how such effect is present in the stability of the studied complexes?

**Authors' reply:** We agree with the reviewer that the stability is still expected to exhibit a size dependence at the nm scale. However, we have no means to calculate the surface tension and in extension the Kelvin effect for our systems directly using quantum chemistry. In addition, the supplied reference is on nano-scale water droplets, which would have significant more molecules in them at the same sizes as the clusters we study. Hence, any comment on this aspect would be purely speculative on our side, and we would prefer not to comment further.

(ii) Related to this, it must be noted that the stability can't be studied solely based on additions of acid-base monomer pairs. The gradient Delta(DeltaG) determines the size- and composition-dependent exponential factor in the cluster evaporation rate, or equally in the saturation vapor pressure above the cluster/particle surface, and consequently the size-dependence becomes negligible when Delta(DeltaG) levels off. However, this behavior is not necessarily same for single monomers, as evaporation of molecule pairs is typically rare event compared to evaporation of molecules.

Can the results be expected to be similar when studying the energetics of single-molecule additions instead?

**Authors' reply:** We agree with the reviewer that addition/evaporation of a acid—base pair is a rare event in actual nucleation and is mostly connected with the SA-DMA system. We note that the presented calculations are extremely computationally demanding and as we had to pick a starting point for studying FNPs we chose a 1:1 acid–base ratio, as these compositions have been shown to be more stable in the literature [3] and our previous work. It will be impossible to know whether the exact same results can be expected when studying the energetics of single-molecule additions instead and comments on this would be speculative on our side. We are currently looking into monomer addition/evaporation and hope to report these results in the future. However, it is beyond the scope of the current work. We do agree that it would beneficial to further comment on this aspect. We have added the following sentence to clarify this aspect:

**Added sentence, Page 10, line 229:**
It should be further noted that we only study the FNPs in increments of acid-base pairs in the current work and that the cluster-to-particle transition point might change once data on monomer additions/evaporation becomes available.

4. P15L291: A weight of >30% for the lowest-energy structure in the ensemble is actually not that high. How would considering the ensemble affect the effective binding free energies (e.g. Partanen et al., J. Phys. Chem. A 120, 8613-8624, 2016)? Or why can it be assumed that the ensemble would not affect them?

**Authors' reply:** Indeed, a weight of 30% would imply that low lying minima could be important and have an effect on the multi-conformer free energies. To check how important the low lying minima are for the free energies we calculated the multi-conformer free energies. As shown in table

S2 in the SI including multiple configurations does not significantly lower the effective binding free energies (< 1 kcal/mol). Hence, the effect of multiple configurations is expected to have a minor effect on our calculated free energies. To comment on this aspect, we have added the following sentence to the manuscript:

**Page 15, Line 318:**
Furthermore, by comparing the binding free energies of the lowest-energy configurations in Table S1 with the multi-conformer binding free energies in Table S2, it is evident that including all identified configurations (filtered by RMSD with threshold of 0.4 Å) reduces the effective binding free energies by less than 1 kcal/mol.

5. P10L226: Is the use of self-consistent distribution necessary? Does this artificially change the differences between the binding free energies of different clusters, i.e. Delta(DeltaG)? I'm not convinced that the monomer free energies need to be forced to zero, as the important quantity of interest when studying the growth thermodynamics and nucleation barriers are the differences Delta(DeltaG), not the absolute energies DeltaG.

**Authors' reply:** We agree with the reviewer that the self-consistent distribution is not necessary in our case, as it does not affect the result. However, from a theoretical point of view it is a lot more satisfying, as it sets the free energies of the monomers to zero, instead of an artificial value. To clarify that the choice of using the self-consistent distribution does not change the results, we have added the following:

**Page 10, Line 236:**
At given conditions the "actual" binding free energies are calculated by [4]: $\Delta G_{\text{bind}}(\vec{p}) = \Delta G^{\circ}_{\text{bind}} + RT(1 - \frac{1}{n}) \sum_i \ln(\frac{p_i}{p_{\text{ref}}})$. where $p_{\text{ref}}$ corresponds to a reference pressure (1 atm) and $p_i$ represents precursor (which in our discussion is monomer) partial pressures. This equation differs from previous work on cluster formation at "actual" conditions as these incorrectly generalized the unimolecular nucleation equation. Thus, the equation satisfies self-consistency also for multi–component systems, i.e., having zero free energies for all precursors. We testing the different formulations of the actual free energies at given conditions and found no deviations between the calculated free energies in the current work.

Minor comments:

P1L11 and other such occurrences: the cluster sizes are here denoted as the total number of monomers, i.e. including both acid and base molecules. For clarity and since only 1:1 acid:base compositions are studied, it could be better to use the number of acid-base pairs.

**Authors' reply:** The choice of using the total number of monomers is deliberately done as we are currently also working on the cluster compositions with ±1 acid or base. Hence, the notation is applied for consistency with our future work. Thus, we would prefer to keep the notation.

P1L14: Presumably the SA-AM and SA-MA clusters can grow also beyond 1.8 nm, even if it's not addressed in the present study?

**Authors' reply:** We agree with the reviewer that it is most likely possible to grow beyond these sizes. Previously we used the term "grow to" as a caution as we do not have data of clusters larger than this size. We have rephrased the sentence accordingly:

**Modified sentence, Page 1, Line 14:**
**From:**
For instance, at low temperature (278.15 K) and high precursor concentration (AM = 10 ppb and MA = 10 ppt) the SA–AM and SA–MA systems can form clusters that grow to ∼1.8 nm sizes.

**To:**
For instance, at low temperature (278.15 K) and high precursor concentration (AM = 10 ppb and MA = 10 ppt) the SA–AM and SA–MA systems can form clusters that grow to and likely beyond ∼1.8 nm sizes.

P1L16: SA-TMA clusters are here found to not grow to larger sizes, but could they do this by condensation by some other compounds?

**Authors' reply:** Indeed, other vapours could potentially condense onto the SA-TMA clusters. Without further studies it is difficult to speculate which compounds that might be. Never-the-less, we agree that this should be further clarified in the manuscript. We have modified the following sentences:

**Modified sentence, page 1, Line 15:**
**From:**
... and the SA-TMA system can only form small clusters, that are unable to grow to larger sizes.

**To:**
... and the SA–TMA system (1:1 acid–base ratio) can only form small clusters, that are unable to grow to larger sizes.

**Page 15, Line 338:**
Hence, under realistic experiment conditions, the presence of water and other species might enable the growth of SA–TMA clusters.

P1L18: Should the statement "aerosol-cloud interactions remain the largest uncertainty in global climate modelling" be rather "aerosol-cloud interactions remain the largest uncertainty in global radiative forcing"?

**Authors' reply:** We agree with the reviewer that this should be further specified. The following sentence has been rephrased accordingly:

**Page 1, Line 18:**
The recent 2023 IPCC report verifies that aerosol-cloud interactions remain the largest uncertainty in global radiative forcing.

P1L23: A "measurable" particle is said to be of approximately 2 nm in diameter: while this is true for many standard atmospheric measurement sites, it must be noted that particles and clusters down to ca. 1 nm sizes or even beyond can be measured by both mass spectrometer and condensation particle counter techniques.

**Authors' reply:** The reviewer is entirely correct that particles and clusters with sizes down to 1 nm can technically be measured. However, there are many challenges in such measurements. For instance, fragmentation of the measured clusters imply that the measured cluster is not necessarily the one that was formed. In addition, particle size magnifiers (PSMs) have poor counting efficiency. We agree that the phrasing "measurable" particle has been used a bit loosely here and have changed the sentence to reflect this:

**Modified sentence, page 1, Line 23:**
**From:**
Under the premise that these clusters do not evaporate or are lost due to coagulation with existing particles, they can grow to larger sizes, eventually becoming a measurable aerosol particle of roughly 2 nm in diameter.

**To:**
Under the premise that these clusters do not evaporate or are lost due to coagulation with existing particles, they can grow to larger sizes, eventually becoming aerosol particles of over 2 nm in diameter, which is the detection limit of many standard atmospheric measurement sites. Smaller particles require mass spectrometer and particle size magnifier techniques.

P10L221-222: The statement about cluster thermochemistry being dependent on the concentration of the clustering monomers should be clarified. What does this exactly refer to? The relative distribution of different cluster compositions, as well as the driving force of condensation/clustering, are indeed dependent on vapor concentrations, but the thermochemical properties of given compositions are not.

**Authors' reply:** We agree with the reviewer that the original description regarding binding free energies is perhaps too vague. The statement was aimed at the discussion of the cluster stability under given conditions, which are certainly depending on monomer vapour pressure. We have rephrased the sentence to the following:

**Page 10, Line 228:**
It should be mentioned that the onset of FNPs is highly dependent on the cluster composition, temperature, and concentration of the clustering monomers.

P10L232: It can be noted that in non-polluted environments, the "low-concentration" limit of 1 ppt for amines can actually be somewhat high, and the concentrations are likely lower outside the vicinity of amine sources. 1 ppt is of the order of $>$1e7 cm$^{-3}$, which can be ca. an order of magnitude higher than the concentration of sulfuric acid in many environments (of the order of 1e6 cm$^{-3}$). Assuming that both SA and amines condense efficiently on aerosol, maintaining an amine concentration of $>$1e7 cm$^{-3}$ would require a constant amine source.

**Authors' reply:** Indeed, we are aware that the applied values for amines may be higher than what is found in non-polluted environments. We chose the values based on observations, which we have compiled in a recent review [5]. However, we agree that it should be further noted in which environments the chosen values are valid. We have added the following sentence to clarify:

**Page 10, Line 248:**
We note that the chosen values of the amines should be viewed as an upper limit in non-polluted environments. Similarly, the concentration of SA could also easily exceed $10^6$ molecules cm$^{-3}$ in many environments.

P12L237: It should be noted that in general, nucleation barriers can't be determined based on only 1:1 compositions, as barriers might be involved in additions of single molecules.

**Authors' reply:** We completely agree with the reviewer that the actual nucleation barriers cannot be determined based on only the 1:1 ratio. Unfortunately, we do not have data on the condensation/evaporation of single molecules (i.e. $\pm$ acid/base) yet. However, this is ongoing work and will be documented in the near future. On the other hand, in most SA–Base system the 1:1 ratio is the most stable composition and if the 1:1 ratio does indeed show a barrier, then the $\pm$ acid/base systems will have to cross that as well. Never-the-less, to clarify we have added the following sentence to the manuscript:

**Sentence added, page 12, Line 256:**
It should be mentioned that the "actual" nucleation barrier of the system can not be determined based solely on the 1:1 acid-to-base ratio composition. Monomer condensation could potentially lead to a change in the barrier. However, in most SA–base systems the 1:1 ratio is the most stable composition and if the 1:1 ratio does indeed show a barrier, then the $\pm1$ acid/base molecule systems will have to cross that as well.

Please review the citations as some are not written correctly, such as "IPC" and "Sipilää et al." on P1.

**Authors' reply:** We apologize for the misspelling in our reference and the format issue of the citation. This has been corrected in the revised manuscript.

**References**

[1] H. Wu, M. Engsvang, Y. Knattrup, J. Kubečka, and J. Elm. Improved Configurational Sampling Protocol for Large Atmospheric Molecular Clusters. *ACS Omega*, 8(47):45065–45077, 2023.

[2] Y. Knattrup, J. Kubečka, H. Wu, F. Jensen, and J. Elm. Reparameterization of GFN1-xTB for Atmospheric Molecular Clusters: Applications to Multi-Acid–Multi-Base Systems. *RSC Adv.*, 14:20048–20055, 2024.

[3] T. Olenius, O. Kupiainen-Määttä, I. K. Ortega, and H. Vehkamäki. Free Energy Barrier in the Growth of Sulfuric Acid–Ammonia and Sulfuric Acid–Dimethylamine Clusters. *J. Chem. Phys.*, 139(8):084312, 2013.

[4] R. Halonen. A Consistent Formation Free Energy definition for Multicomponent Clusters in Quantum Thermochemistry. *J. Aerosol Sci.*, 162:105974, 2022.

[5] M. Engsvang, H. Wu, Y. Knattrup, J. Kubečka, A. Buchgraitz Jensen, and J. Elm. Quantum Chemical Modeling of Atmospheric Molecular Clusters Involving Inorganic Acids and Methanesulfonic Acid. *Chem. Phys. Rev.*, 4(3):084312, 2023.

---

## Referee Report (RR1)

I'd like to thank the authors for the replies to mine and other reviewers' comments, and the clarifications made in the manuscript.

I would like to ask further clarifications related to the reply to my previous comment 2: The authors reply that they did mean that the discussed quantity in relation to Fig. 4b is the actual gradient $\Delta(\Delta G)$, which I suggested to use.

I apologize for being unclear; what I actually meant was to study the gradient $\Delta(\Delta G)$ without dividing the $\Delta G$ data by $m$. This is because the $1/m$ factor will always lead to a similar curve that first changes steeply and then levels off—approximately at $m$ values of ca. 15-20 molecules—*regardless of the exact chemical properties*.

That is, when you scale the numbers in Fig. 4a by $1/m$, the shape of the curve is determined by this factor, so you are simply bending the graphs in Fig. 4a:

[Figure]

One can think of e.g. an example case of $\Delta G$ data where each energy contribution $\Delta(\Delta G)$ upon addition of monomer pair is approximately same, i.e. the gradient is $\Delta(\Delta G)$ constant, and there isn't any size-dependent transition. However, scaling this data by $1/m$ naturally results in the shape in the plot above:

[Figure]

It clearly doesn't sound reasonable to interpret this as a physical or chemical transition from clusters to particles. (Note that just to illustrate the risk in such interpretation, this example is qualitatively similar to Fig. 4.)

I'm unsure of the physical interpretation of the $\Delta G/m$ quantity. It is said to be "average binding free energy contribution in the clusters". Generally, the composition- and size-dependent contribution of monomers (or monomer pairs) to the binding free energy is described by the change upon addition of molecules, i.e. $\Delta(\Delta G)$; in contrast, the average $\Delta G/m$ doesn't contain any information of these

contributions. For the data in Fig. 4a, the actual size-dependent contributions $\Delta(\Delta G)/\Delta m$ look approximately like this:

[Figure]

This is quite different from the curves in Fig. 4b.

Therefore, I would ask the authors to at least add the following:

1. Clarify why diving the binding free energy by the total number of molecules is expected to give information on the cluster-to-particle transition, and what the physical meaning of this quantity is. Preferably give references, if possible, since I think that this might not be clear to all readers.

2. Please show and/or discuss how $\Delta G/m$ relates to the actual size-dependent monomer (pair) contribution $\Delta(\Delta G)/\Delta m$, and why $\Delta G/m$ should be used instead of e.g. $\Delta(\Delta G)/\Delta m$. I still recommend to show also $\Delta(\Delta G)$ (it's okay if it doesn't look as perfect as the smooth $\Delta G/m$ curves).

3. Explain that the general shape and leveling off of the $\Delta G/m$ curves is primarily not related to the chemical properties or composition, but simply follows from the $1/m$ factor which is same for all chemistries.

---

## Author Response (AR2)

We would like to thank the second anonymous reviewer for clarifying comments. In the revised version of the manuscript we have carefully taken each comment into account and believe it has improved the study. The reviewer's comments have been reproduced in blue text below, followed by our point-by-point replies.

**Anonymous Referee 2**

I'd like to thank the authors for the replies to mine and other reviewers' comments, and the clarifications made in the manuscript.

I would like to ask further clarifications related to the reply to my previous comment 2: The authors reply that they did mean that the discussed quantity in relation to Fig. 4b is the actual gradient $\Delta(\Delta G)$, which I suggested to use.

I apologize for being unclear; what I actually meant was to study the gradient $\Delta(\Delta G)$ without dividing the $\Delta G$ data by m. This is because the 1/m factor will always lead to a similar curve that first changes steeply and then levels off—approximately at m values of ca. 15-20 molecules—regardless of the exact chemical properties.

That is, when you scale the numbers in Fig. 4a by 1/m, the shape of the curve is determined by this factor, so you are simply bending the graphs in Fig. 4a:

[Figure]

One can think of e.g. an example case of $\Delta G$ data where each energy contribution $\Delta(\Delta G)$ upon addition of monomer pair is approximately same, i.e. the gradient is $\Delta(\Delta G)$ constant, and there isn't any size-dependent transition. However, scaling this data by 1/m naturally results in the shape in the plot above:

[Figure]

It clearly doesn't sound reasonable to interpret this as a physical or chemical transition from clusters to particles. (Note that just to illustrate the risk in such interpretation, this example is qualitatively similar to Fig. 4.)

I'm unsure of the physical interpretation of the $\Delta G/m$ quantity. It is said to be "average binding free energy contribution in the clusters". Generally, the composition- and size-dependent contribution of monomers (or monomer pairs) to the binding free energy is described by the change upon addition of molecules, i.e. $\Delta(\Delta G)$

; in contrast, the average $\Delta G/m$ doesn't contain any information of these contributions. For the data in Fig. 4a, the actual size-dependent contributions $\Delta(\Delta G)/\Delta m$ look approximately like this:

[Figure]

This is quite different from the curves in Fig. 4b.

Therefore, I would ask the authors to at least add the following:

1. Clarify why diving the binding free energy by the total number of molecules is expected to give information on the cluster-to-particle transition, and what the physical meaning of this quantity is. Preferably give references, if possible, since I think that this might not be clear to all readers.

**Authors' reply:** We agree with the reviewer that it would be beneficial to further discuss the physical meaning of the size-averaged binding free energy $\Delta G_{\mathrm{bind}}/m$. Although references on this property are scarce, apart from our own previous work, the recent work by Sindel et al. [1] used a similar definition to study $TiO_2$ clustering, leading to a convergence in the size-averaged binding free energies toward the formation free energy of the bulk crystal.

The physical interpretation of the size-averaged binding free energy is the binding free energy per monomer in the cluster. This quantity provides insight into the average binding properties of the cluster and offers inferred evidence about the thermochemistry associated with monomer addition. By analyzing how the average binding free energy changes with cluster size, we can understand the stabilization processes occurring during cluster growth.

For example, consider the difference in average binding free energy between a $(SA)_{99}(AM)_{99}$ cluster and a $(SA)_{100}(AM)_{100}$ cluster. In such large clusters, adding one extra acid-base pair results in minimal molecular rearrangement, and thus the average binding free energy remains largely unchanged. This behavior is analogous to condensation thermodynamics. In contrast, adding one $(SA)_1(AM)_1$ pair to form a $(SA)_2(AM)_2$ cluster results a huge drop in the average free energy, as the addition causes a large stabilization at such small cluster sizes by a significant molecular rearrangement. Once the first encapsulated ion is present, further stabilization corresponds to the addition of monomers to an existing solvent shell. This illustrates a gradual transition from single molecules to clusters, highlighting the informative nature of the average binding free energy in this context.

To further elaborate on this aspect, we have added a small section to the methods describing the size-averaged binding free energies in more detail:

**Added paragraph, page 4, line 110:**

We calculate the binding free energies as the free energy of the cluster with respect to its individual monomers:

$$\Delta G_{\text{bind}} = G_{\text{cluster}} - \sum_i G_{\text{monomers},i} \tag{1}$$

We also calculate the size averaged binding free energies ($\Delta G_{\text{bind}}/m$) of the clusters as the physical interpretation is the binding free energy contribution per monomer in the cluster. This quantity provides insight into the average binding properties of the cluster and offers inferred evidence about the thermochemistry associated with monomer addition. Analyzing how the average binding free energy changes with cluster size will present to us the stabilization processes occurring during cluster growth. A recent work by [1] used a similar definition to study $TiO_2$ clustering, leading to a convergence in the size-averaged binding free energies toward the formation free energy of the bulk crystal. For example, consider the difference in average binding free energy between a $(SA)_{99}(AM)_{99}$ cluster and a $(SA)_{100}(AM)_{100}$ cluster. In such large clusters, adding one extra acid-base pair results in minimal molecular rearrangement, and thus the average binding free energy remains largely unchanged. This behavior is analogous to condensation thermodynamics. In contrast, adding one $(SA)_1(AM)_1$ pair to form a $(SA)_2(AM)_2$ cluster results a huge drop in the average free energy, as the addition causes a large stabilization at such small cluster sizes by a significant molecular rearrangement.

We also agree with the reviewer that the leveling off in the $\Delta G_{\text{bind}}/m$ versus $m$ curve alone is not sufficient to classify the transition from clusters to particles. A monomer within the cluster being fully surrounded by adjacent monomers would very likely result a significant difference of binding free energy comparing to the configurations without encapsulated monomer, as the monomer can fully interact with its surroundings. We applied convex-hull analysis to verify that an ion is actually fully encapsulated. In the regime of $4 < m < 16$, we found that the size-averaged binding free energy does not change significantly as a function of $m$. We preliminarily consider this regime to represent the cluster-to-particle transition. In addition, when combined with the structural changes identified using the convex hull method, it provides additional inferred evidence supporting this transition. We believe this is well captured in the following sentence on page 1, line 7:

*"We define the onset of FNPs as when one or more ions are fully solvated inside the cluster and when the gradient of the size-averaged binding free energy approaches zero"*

2. Please show and/or discuss how $\Delta G/m$ relates to the actual size-dependent monomer (pair) contribution $\Delta(\Delta G)/\Delta m$, and why $\Delta G/m$ should be used instead of e.g. $\Delta(\Delta G)/\Delta m$. I still recommend to show also $\Delta(\Delta G)$ (it's okay if it doesn't look as perfect as the smooth $\Delta G/m$ curves).

**Authors' reply:** We agree with the reviewer that the appropriate quantity for estimating stability is the addition free energy $\Delta G_{\text{add}}$, as it is related to the evaporation rate. However, our data is limited to clusters with a 1:1 acid-to-base ratio, which means our initial figure only depicted the addition of an acid-base pair. Since dimer evaporation is relatively rare in SA–base clusters, interpreting such a figure in terms of overall stability is challenging. Mathematically, taking the end point (at $m = 30$) of the size averaged binding free energy ($\Delta G/m$) corresponds to taking the average of the addition energetics. We acknowledge that discussion about the mathematical relation between $\Delta(\Delta G)/\Delta m$ and $\Delta G/m$ would be beneficial.

To address this, we have expanded Figure 4 with the acid–base pair addition $\Delta G_{\text{add}}$ free energies. Instead of $\Delta(\Delta G)/\Delta m$, we opted for $\Delta G_{\text{add}}$ since it directly represents the addition free energies of a pair of acid–base. As illustrated in the figure, the addition free energies fluctuate around the convergence values (dotted lines). A paragraph was added to describe the relation between $\Delta(\Delta G)/\Delta m$ and $\Delta G/m$.

[Figure]

**Figure 1:** *Left:* Total binding free energies of $(SA)_n(AM/MA/DMA/TMA)_n$ clusters, $n = 1\text{--}15$. *Middle:* Size-averaged binding free energy contribution in the clusters. *Right:* Acid–base pair addition free energies $\Delta G_{\text{add}}$ of the four SA–Base clusters. (The dot line indicates the values of $2 \times \Delta G_{\text{bind}}/m$ with $m = 30$, the values were multiplied by two as we present pair wise addition here.) Data of SA-AM/DMA are from our prior work [2], with additional sampling carried out in this work.

**Added paragraph, page 10, Line 237:**
As shown in the right panel of Figure 1, the acid–base pair addition free energies $\Delta G_{\text{add}}$ fluctuate around the convergence value of the size-averaged binding free energy per monomer, $\Delta G_{\text{bind}}/m$ (indicated by the dotted lines). Since the addition free energy $\Delta G_{\text{add}}$ is the accurate quantity for estimating cluster stability, the relation between addition free energy $\Delta G_{\text{add}}$ and size–averaged binding free energy $\Delta G_{\text{bind}}/m$ is significant. Mathematically, taking the endpoint at $m$ of the size-averaged binding free energy $\Delta G_{\text{bind}}/m$ corresponds to calculating the average of the addition free energies up to cluster size $m$. This approach helps determine the cluster size $m$ at which fluctuations in $\Delta G_{\text{bind}}$ become negligible relative to the entire system. Furthermore, the convergence value of $\Delta G_{\text{bind}}/m$ can be used to estimate the free energies of acid–base pair additions to large clusters.

3. Explain that the general shape and leveling off of the $\Delta G/m$ curves is primarily not related to the chemical properties or composition, but simply follows from the $1/m$ factor which is same for all chemistries.

**Authors' reply:** Indeed, while multiplying a linear $\Delta G_{\text{bind}}$ curve by $1/m$ will always produce a bend, the linearity of the binding free energy $\Delta G_{\text{bind}}$ as a function of $m$ is dependent on the chemical composition of the system. In our case, we study strongly bound systems and observe a linear trend. However, this is not always given *a priori*. For instance, in the previous work done by Myllys *et. al.*[3] (See their Figure 1, 6 and 10), the $\Delta G_{\text{bind}}$ of $(SA)_1(Base)_1(H_2O)_n$ is plotted against the number of water molecules $n$ (where $n = m - 2$, with $m$ being the total monomer count, including one SA and one base molecules). In all these cases the free energy is not simply linearly decreasing as these interactions are quite weak. For instance, hydration of bases even gives an increasing free energy as a function of water molecules (Ref [3], Figure 1). This further supports the idea that the behavior of $\Delta G_{\text{bind}}/m$ is system-specific and underscores the importance of considering both energetic and structural factors when interpreting these curves. To further clarify this aspect, we have added the following sentence when first discussing the linear trend in the binding free energies.

**Added paragraph, page 8, line 201:**
It should be noted that the linearly decreasing trend is dependent on the cluster composition and requires strongly binding clusters. For instance, in the previous work done by [3] (As shown in Figure 1, 6 and 10 of their paper), the $\Delta G_{\text{bind}}$ of $(SA)_1(Base)_1(H_2O)_n$ is plotted against the number of water molecules $n$ (where $n = m - 2$, with $m$ being the total monomer count, including one SA and one base molecules). In all these cases the free energy is not simply linearly decreasing as the inter–molecular interactions are quite weak. For instance, hydration of bases even gives an increasing free energy as a function of water molecules (see [3], Figure 1).

**References**

[1] J. P. Sindel, D. Gobrecht, Ch. Helling, and L. Decin. Revisiting Fundamental Properties of TiO2 Nanoclusters as Condensation Seeds in Astrophysical Environments. *Astron. Astrophys*, 668:A35, 2022.

[2] H. Wu, M. Engsvang, Y. Knattrup, J. Kubečka, and J. Elm. Improved Configurational Sampling Protocol for Large Atmospheric Molecular Clusters. *ACS Omega*, 8(47):45065–45077, 2023.

[3] N. Myllys, D. Myers, S. Chee, and J. N. Smith. Molecular Properties Affecting the Hydration of Acid–base Clusters. *Physical Chemistry Chemical Physics*, 23(23):13106–13114, 2021.